# NMR-Based Metabolomics Demonstrates a Metabolic Change during Early Developmental Stages from Healthy Infants to Young Children

**DOI:** 10.3390/metabo13030445

**Published:** 2023-03-18

**Authors:** Liana Bastos Freitas-Fernandes, Gabriela Pereira Fontes, Aline dos Santos Letieri, Ana Paula Valente, Ivete Pomarico Ribeiro de Souza, Tatiana Kelly da Silva Fidalgo

**Affiliations:** 1National Center for Nuclear Magnetic Resonance, Medical Biochemistry, Universidade Federal do Rio de Janeiro, Rio de Janeiro 21941-902, RJ, Brazil; 2Department of Pediatric Dentistry and Orthodontics, School of Dentistry, Universidade Federal do Rio de Janeiro, Rio de Janeiro 21941-590, RJ, Brazil; 3Department of Preventive and Community Dentistry, School of Dentistry, Universidade do Estado do Rio de Janeiro, Rio de Janeiro 20551-030, RJ, Brazil

**Keywords:** metabolome, metabolites, saliva, infant, children, nuclear magnetic resonance

## Abstract

The present study aims to identify the salivary metabolic profile of healthy infants and young children, and to correlate this with age, salivary gland maturation, and dentition. Forty-eight children were selected after clinical evaluation in which all intraoral structures were examined. Total unstimulated saliva was collected, and salivary metabolites were analyzed by 1H Nuclear Magnetic Resonance (NMR) at 25 °C. Partial least squares discriminant analysis (PLS-DA), orthogonal PLS-DA (O-PLS-DA), and univariate analysis were used, adopting a 95% confidence interval. The study showed a distinct salivary metabolomic profile related to age and developmental phase. The saliva of children in the pre-eruption teeth period showed a different metabolite profile than that of children after the eruption. However, more evident changes were observed in the saliva profile of children older than 30 months. Alanine, choline, ethanol, lactate, and sugar region were found in higher levels in the saliva of patients before 30 months old. Acetate, N-acetyl sugar, butyrate, caproate, creatinine, leucine, phenylalanine, propionate, valine, succinate, and valerate were found to be more abundant in the saliva of children after 30 months old. The saliva profile is a result of changes in age and dental eruption, and these findings can be useful for monitoring the physiological changes that occur in infancy.

## 1. Introduction

Saliva is a complex biofluid containing numerous biological compounds which has high diagnostic potential for clinical outcomes [1,2,3,4,5,6]. This biofluid contains proteins, lipids, and low-molecular-weight metabolites that are products of cellular physiological reactions, and it performs an essential function in the oral environment [7]. The metabolic profile of saliva reflects the contribution of endogenous oral and systemic metabolism, as well as exogenous components such as oral flora and dietary products [8,9]. The metabolomic approach can determine the signatures of several local and systemic conditions, such as diabetes, periodontal disease, Sjogren syndrome, etc. [10,11,12,13]. To this end, this group characterized the profile of salivary metabolites in healthy children of different ages, including infants [14], with and without caries [8,15]. This study also determined the metabolites that are produced by oral microbiota metabolism [9,16] which characterize systemic diseases in children, as well as type I diabetes [12] and renal failure [11,13]. To allow for full saliva-monitoring potential, physiological modifiers such as oral microbiota metabolism products, diurnal variation, tooth eruption, diet, and age should be understood. In infancy, specific events such as salivary gland development and tooth eruption can modify the salivary profile. Salivary gland development begins in the sixth to eighth embryonic week and involves interactions between the epithelium and the underlying mesenchyme to form the functional part of the tissue. The parotid gland is the first to develop, and a hollow tube or duct of major salivary glands is formed in the sixth month of life, dramatically increasing the salivary flow rate [17]. In this period, the sixth month of life, the first primary teeth erupt. The salivary metabolomic profiles before and after this event are thought to be distinct. In the 30th month of life, another critical event occurs during stomatognathic system maturation, where the primary dentition is completely established, increasing the surface area for microorganism colonization and changing the local microbiota population [18,19].

Besides these developmental changes, during the first years of life there is a dietary transition and a significant increase in the diversity of microbial species that colonize the oral cavity [20,21,22], which may also be responsible for altering salivary composition. Neyraud et al. (2020) found metabolic profile changes between three and fifteen months of age, with increased 2-aminobenzoic acid, alanine and phenylalanine, hydroxybutyric acid, and acetoacetic acid levels. Glucose, maltose, lactose, and choline decreased over the first year [23].

There is limited knowledge related to the saliva profile of infants in the first months of life, mainly due to the difficulty of obtaining samples during this period. The literature reports that during infancy, some transformation occurs in the protein composition of saliva. Specifically, between three and six months of age there are higher levels of b-2 microglobulin and S-type cystatins [24,25]. The expression of mucins MG1 (MUC5B) and MG2 (MUC7) changes during the first year of life, with MG2 showing higher levels at the beginning of the first year and MG1 showing higher levels at the end [24]. Different glands produce MGs; MG1 is expressed in submandibular, sublingual, and some minor salivary gland cells, while MG2 is expressed in the submandibular gland [26]. Ruhl et al. (2005) already observed that most salivary proteins are expressed as early as the first month of life [24]. Sonesson et al. (2011) also demonstrated that some salivary proteins, such as mucin MG2 (MUC7) and IgA, increase with age, after comparing saliva from 3-year-old children, 14-year-old adolescents, and 25-year-old adults [27]. Although some information is present in the literature related to the protein content in infant saliva, limited knowledge is available concerning their levels of low-molecular-weight metabolites.

Therefore, the present study aims to characterize the salivary metabolomic profile during the early developmental stages of healthy infants and young children, using Nuclear Magnetic Resonance.

## 2. Experimental Design

### 2.1. Research Subjects

Forty-eight systemically healthy infants and children from zero to five years old, without oral lesions or soft tissue lesions, were included in this study. Children with erupted teeth should be in primary dentition, without tooth decay, to be included. All parents of included subjects signed a Free and Informed Consent Form (TCLE) before participation in this study. Data and sample collection were carried out at the Pediatric Dentistry Clinic of the Federal University of Rio de Janeiro, UFRJ. Those responsible filled out a form developed for the study with personal data, anamnesis, health information, and the participants’ hygiene and dietary habits. Then, a trained professional performed the clinical examination of all the children’s intraoral structures, recording the data on a clinical examination form. For the intraoral examination, a mouth mirror, an explorer probe, and a disposable tongue retractor were used. The study protocol and the use of human material were authorized by the Research Ethics Committee (Number 4.712.999).

### 2.2. Collection and Storage of Saliva Samples

Samples were collected in the period between 8:00 am and 10:00 am, due to the circadian saliva cycle [28]. Mothers were asked to refrain from feeding 15 min before saliva collection. Infants’ oral mucosa were cleaned using a gauze moistened with filtered water 5 min prior to saliva collection. Children older than 2 years were asked to suspend oral activities for 1 h prior to saliva collection. Unstimulated total saliva of the participants (0.5 mL) was collected using an automatic pipette with sterile tips. It was then stored in sterile plastic tubes (Eppendorf-TM) on ice and centrifuged for 1 h at 10,000× *g* and 4 °C in the laboratory of the Faculty of Dentistry UFRJ. The supernatant was stored at −80 °C until the moment of analysis.

### 2.3. Sample Preparation for Nuclear Magnetic Resonance (NMR), Data Acquisition and Analysis

The samples were prepared and analyzed in a 400 MHz spectrometer at the National Center of Nuclear Magnetic Resonance by Jiri Jonas at Universidade Federal do Rio de Janeiro (CENABIO/UFRJ), who performed hydrogen NMR (^1^H) of the solutions using a Carl-Purcell-Meiboom-Gill (CPMG) pulse sequence.

A mixture of 540 μL of saliva supernatant from the respective subjects plus 60 μL of pH 7 phosphate buffer was prepared, containing 99.8% (to provide a field-frequency lock) and 20 µM of 4,4-dimethyl-4-silapentane-1-sulfonic acid (DSS) (for chemical shift referencing, δ = 0.00 ppm) [14,29]. The spectra acquisitions (0–12 ppm) were performed in a standardized way, with a pre-established receiver gain and number of scans (1024 scans) and a calculated pulse time and signal/noise ratio. The CPMG pulse sequence was used at 298 K (25 °C).

Data were subjected to correction and base alignment using the software Topspin 4.1.4 (Bruker Biospin program, Rheinstetten, Germany). ^1^H-^1^H total correlation (TOCSY) experiments were also conducted, with acquisition parameters of 2048 × 256 complex points, a spectral width of 12,019 Hz in each dimension, and a mixing time of 70 ms [8,14]. Spectra and spectral regions that could not be corrected for phase and baseline were excluded from the analyses. The marking strategy included the use of the Human Metabolome database (http://www.hmdb.ca/, accessed on 24 June 2022 ) and previous markings in the literature [8,14,29]. TOCSY experiments were used to confirm the assignments. Thus, the relative abundances of metabolites in the study groups were obtained and subjected to statistical analysis.

### 2.4. Statistical Analysis

The statistical program SPSS 20.0 (IBM, Chicago, IL, USA) was used to store and analyze the data obtained, apply normality tests, and stipulate a level of statistical significance of 95% to be used.

The spectra referring to the metabolomics data were submitted to the AMIX program (Bruker Biospin, Rheinstetten, Germany) for data extraction. Initially, the spectra were divided into 0.03 ppm buckets, and the water region was removed (4.5–5.5 ppm) to eliminate interference in the spectrum. Although the amount of fluid components does not change, there is evidence of variation in the ratio of water to components. Therefore, since doing so does not interfere with multivariate analysis, the values of each peak of the spectra were normalized, divided by the sum of the signal intensities, and then subjected to the Pareto scaling method [30] before applying the multivariate analysis.

The generated data matrix, containing the peaks and integrals of the buckets’ areas, was subjected to multivariate analysis using the Metaboanalyst 5.0 program (www.metaboanalyst.ca, accessed on 24 June 2022). Metaboanalyst 5.0 was also used to obtain the predictive performance of the models; each model was evaluated for Q^2^, R^2^, and accuracy (ACC) for cross-validation purposes [31]. Principal Component Analysis (PCA), discriminant analysis with partial least squares method (PLS-DA), and discriminant analysis by orthogonal projections to partial least squares (O-PLS-DA) were also applied. Analysis of the VIP scores based on PLS-DA determined which salivary components contributed most to the differences between groups, related to the presence or absence of teeth [32]. A hierarchical clustering by dendrogram analysis was performed, which considered participants’ ages (< or >30 months) using euclide distance measurement and the Ward clustering algorithm (Metaboanalyst 3.0). For univariate analysis, data were submitted to the t test, adopting a 95% confidence interval. A metabolic pathway analysis was performed using the compound names of the metabolites that resulted from univariate and VIP score analysis as the input variables; the pathway library used was the KEGG human metabolic pathways database (Metaboanalyst 3.0). The confidence interval was set to 95%. To improve transparency, data extraction is available in the Open Science Framework repository (https://doi.org/10.17605/OSF.IO/VFXRA, accessed on 24 June 2022).

## 3. Results

This study was able to characterize the salivary profile of infants and children in different developmental stages, ranging from 16 to 60 months of life. The group containing infants without teeth consisted of twenty-six subjects (fifteen girls) with a mean age of 2.3 ± 1.9 months. The group of children with teeth consisted of twenty-two subjects (sixteen girls) with a mean age of 44.9 ± 18.1 months.

Figure 1 shows the representative ^1^H NMR spectrum (0.50–4.50 ppm) of children’s saliva after primary teeth eruption (at 20 months).

Multivariate analyses were performed to assess metabolite differences between groups. The PCA (Appendix A) was performed using age and dentition as identifiers (PC1 × PC2 and PC1 × PC3), and age presented better separation (Appendix A) compared to tooth eruption (Appendix A). The PLS-DA and O-PLS-DA of children before and after dental eruption (Figure 2A,B), as well as differences in samples from children before and after 30 months old (Figure 2D,E), show separation between groups; the model (Figure 2A) presented satisfactory accuracy and prediction. Of the three main principal components for dental eruption, ACC = 0.86, R^2^ = 0.74, and Q^2^ = 0.48. For the first two main components, PC1 and PC2, the PLS-DA presented a variability of 35% (Figure 2A) and the O-PLS-DA a variability of 33.2% (Figure 2B). For the 30-month-old separation, the model (Figure 2D) also presented satisfactory accuracy and prediction with regard to the three main components: ACC =0.95, R^2^ = 0.86, and Q^2^ = 0.71. For the first two main principal components, PC1 and PC2, the PLS-DA presented a variability of 35.2% (Figure 2D) and the O-PLS-DA a variability of 32% (Figure 2E). Figure 2C,F shows the VIP scores that were generated from the multivariate analysis, thus indicating the metabolites most responsible for the differences between groups.

The highest VIP score demonstrates that the metabolite was the most important for the separation between groups. Thus, acetate, N-acetyl sugar, valine, valerate, and butyrate were present in higher amounts in the saliva of patients after primary teeth eruption compared to saliva before teeth eruption. On the other hand, lactate, glucose, and sugar region were found in greater amounts in the saliva of patients before teeth eruption. Acetate, N-acetyl sugar, and aminobutyrate were present in higher levels in the saliva of patients over 30 months old than in the saliva of those under 30 months old. Lactate, glycerol, glucose, and sugar region were found in greater amounts in the saliva of patients before 30 months old.

Table 1 shows salivary metabolites from saliva before and after 30 months of age, using the chemical shift, peak intensity variation, and *p* values. The intensities of saliva metabolites, presented as mean (arbitrary units), confidence interval, spectrum region (ppm), and statistical analysis of multivariate analysis^1^ and univariate analysis^2^, were used. Univariate analysis also showed differences between the groups. Thus alanine, choline, ethanol, lactate, and sugar region were found in higher amounts in the saliva of patients before 30 months of age compared to saliva of patients after 30 months. Acetate, butyrate, caproate, creatinine, leucine, N-acetyl-sugar, phenylalanine, propionate, succinate, trimethylamine, valerate, and valine were higher after 30 months of age than before. Univariate analysis showed salivary metabolites from saliva before and after teeth eruption and found higher amounts of lactate and sugar region before tooth eruption and more butyrate, propionate, and acetate after.

Figure 3 shows the change in the intensity of normalized peaks, representing the amount of each metabolite in each child’s saliva showing consistent changes over time. These data are in line with the metabolite production during the children’s growth. It is possible to observe that after 30 months of life, creatinine, N-acetyl sugar, propionate, and valine production increased, and ethanol production decreased. Interestingly, while some metabolites show a constant increase or decrease over time, such as choline, alanine, ethanol, and propionate, others show a threshold at close to 30 months of age, such as acetate, creatine, creatinine, leucine and valine. Beyond this threshold, their values increase and scatter. The figure also shows infants without and with teeth in blue and red, respectively.

The dendrogram (Figure 4) demonstrates a distinction between infants < and >30 months of age. In the <30-month-old group, one miss classification (Baby 31) can be observed, as well as two miss classifications (Baby 18 and 25) in the >30-month-old group.

The pathway analysis (Figure 5) considers the metabolites acetate, alanine, butyrate, choline, creatinine, ethanol, formate, glycerol, lactate, leucine, n-acetyl-sugar, phenylalanine, propionate, succinate, sugar region, trimethylamine, valerate, valine, caproic acid, and urea. The top 10 metabolites presented a statistical difference; therefore, the most important pathways were Aminoacyl-tRNA biosynthesis (*p* < 0001); Glycolysis/Gluconeogenesis (*p* = 0.001); Valine, Leucine, and Isoleucine biosynthesis (*p* = 0.00139); Butanoate metabolism (*p* = 0.00209); Pyruvate metabolism (*p* = 0.00757); Propanoate metabolism (*p* = 0.0161); Alanine, Aspartate, and Glutamate metabolism (*p* = 0.0175); Glyoxylate and Dicarboxylate metabolism (*p* = 0.0255); Phenylalanine, Tyrosine, and Tryptophan biosynthesis (*p* = 0.0327); and Valine, Leucine, and Isoleucine degradation (*p* = 0.036).

## 4. Discussion

The present study observed the salivary metabolic profile of healthy infants and young children during different developmental stages. The eruption of teeth is an important event that occurs during the early stages of life, and ^1^H-NMR metabolomics demonstrated different profiles before and after this event. It was also found that a metabolomic shift in saliva occurs after 30 months old, which represents a threshold where events coincide with the completion of dentition, suggesting that this is related to the maturation of salivary glands and the increasing colonization of surfaces by oral microorganisms.

During the first year of life, an infant develops further in association with the natural physiological process of primary dentition eruption, with local and general manifestations [33]. Increased salivary flow is one of the manifestations frequently found in children during the eruption phase of the first primary teeth [33]. There are many physiological modifiers that are difficult to control, which represent limitations to this type of study [34]. Despite this, several factors correlate saliva and its components to the normal growth and development of the child, which may be responsible for the changes observed [35]. Our study showed variation in the salivary metabolites profile related to different pathways, including amino acids and glucose metabolism. The top 10 related metabolic pathways were, respectively, Aminoacyl-tRNA biosynthesis; Glycolysis/Gluconeogenesis; Valine, Leucine, and Isoleucine biosynthesis; Butanoate metabolism; Pyruvate metabolism; Propanoate metabolism; Alanine, Aspartate, and Glutamate metabolism; Glyoxylate and Dicarboxylate metabolism; Phenylalanine, Tyrosine, and Tryptophan biosynthesis; and Valine, Leucine, and Isoleucine degradation.

More choline was found within the first year of age than in older children, a finding corroborated by previous studies [36]. Choline is a precursor of several molecules, such as acetylcholine and phospholipids, and its decrease within the first year of life has been related to the decrease in milk consumption [37]. In the present study, children younger than 30 months were more likely to use formula than older children, which constitutes a possible explanation for the reduction of choline levels in this studied population.

Changes in the concentration of amino acids in whole saliva can be related to endogenous proteolysis and bacterial metabolic pathways [8,9,14]. The levels of phenylalanine and valine were higher in the saliva samples of patients older than 30 months, which could be related to proteolysis around the teeth eruption site. In addition, the metabolic pathway demonstrated phenylalanine, tyrosine, and tryptophan biosynthesis and valine, leucine, and isoleucine degradation. Barnes et al. (2011) showed that increased levels of phenylalanine and valine are also related to the inflammation process that may be active in teeth eruption [38]. In the present study, no important changes were found in proline or its byproducts that could be related to the proteolysis of mucin or PRP, which could be an indication of the maturation process of the submandibular gland [17].

The present study also demonstrated higher levels of organic acids such as acetate, butyrate, succinate, and propionate with age, especially after tooth eruption. Morzel et al. (2011) described changes in salivary protein profiles in infants between three and six months old related to dietary modification that occurs during the first year [39]. Such changes in salivary proteins may provide substrates and surfaces for oral microbial adhesion and colonization [20,39,40]. Saccharolytic microflora convert sugars to lactic, formic, acetic, succinic, and other organic acids to obtain energy [38]. The concentration of these metabolites has been related to the bacterial load [41]. Fidalgo et al. (2013) used the metabolomic approach to analyze saliva from individuals with primary, mixed, or permanent dentition, from three to twelve years old [14]. This study demonstrated a similar general profile independent of dentition and age; however, some metabolites suggested to be related to oral microorganisms’ metabolites, such as organic acids, were observed in higher levels in permanent teeth. It is suggested that the increase in surface area with tooth eruption favors oral bacteria colonization and bacterial load, resulting in an upregulation of organic acid levels. This was also observed in the current study.

It was found that acetate levels were increased in children with erupted primary teeth compared to toothless children. In previous findings, Fidalgo et al. (2015) demonstrated that this metabolite was higher with permanent dentition and in the saliva of patients with caries [8]. Acetate is found in the saliva produced by the parotid, submandibular, and sublingual glands [42]. This metabolite is a compound formed by bacterial metabolism, present in the dental biofilm matrix. N-acetyl sugar is a microbial product that was also found in greater amounts in children’s saliva after tooth eruption. This metabolite is an amide derivative of the monosaccharides. Butyrate and propionate are metabolites that are related to bacterial metabolism [29,43]. These metabolites were also found in greater amounts in children after dental eruption. Comparing spectra, the saliva samples of patients with gingival inflammation/periodontal disease were characterized by higher levels of acetate, c-aminobutyrate, n-butyrate, succinate, trimethylamine, propionate, and valine than those of healthy patients [44]. Children after teeth eruption are more prone to developing gingival inflammation [44], which is consistent with our results.

Changes were also observed in compounds that may be related to food intake and oral hygiene, such as ethanol and glycerol. A decrease in salivary sugar concentration with age was observed, perhaps because, it is speculated, infants present more frequent feeding than older children. In addition, the feeding of children within the two groups varied between exclusive breastfeeding, use of formulas, and solid foods [25].

Our data showed an increase in creatine and creatinine with age, including, importantly, beyond the threshold of 30 months. Creatine and creatinine are metabolites found in muscle cells, and their increase may be related to the enormous change in weight and height observed during this period of childhood [23]. Creatinine concentration in the blood and urine is a marker of glomerular filtration influenced by several factors and has been correlated with age in previous metabolomics studies [45], therefore corroborating our data. Creatine and creatinine have already been related to age. Barr et al. (2005) studied a population with ages ranging from six to seventy years and demonstrated that urinary creatinine increases between 20 and 29 years and decreases thereafter [46]. Gu et al. (2009) also demonstrated an increase in urinary creatinine levels with age, from youth to adulthood [47].

The circadian cycle also seems to change the metabolic profile [48]. Our study was not able to distinguish this effect. The study’s design attempted to avoid interference by collecting saliva samples at the same time of day, in the morning. Our study aimed to determine the impact of tooth eruption and uncontrolled changes that occur during the first months of life on salivary metabolomic analysis of healthy infants and children. This information could impact studies in different areas such as dentistry, medicine, nutrition, development, and physiology, and it could influence the full assessment and monitoring of different types of diseases, both oral and systemic [49,50].

A limitation of the present study was the cross-sectional study design. Further studies should be conducted, such as a longitudinal study where the same infant would be followed after tooth eruption and after 30 months. Based on the present study, it is suggested that saliva could be another biofluid used for screening programs, especially considering the difficulties in blood collection in this young population. In this sense, metabolomics offers the opportunity for precision medicine that could be highly informative as well as discriminatory for the early detection of several diseases in infants and children. The integration of omics technologies with saliva from infants and children promise the real-time evaluation of their health conditions.

## 5. Conclusions

It can be concluded that the saliva profile of infants and young children varies as a result of changes in age and dentition. This study observed a shift in amino acid, creatine/creatinine, and organic acid metabolites during this early period of life; these metabolites reveal that salivary components change in line with growth and systemic functions in children.

## Figures and Tables

**Figure 1 metabolites-13-00445-f001:**
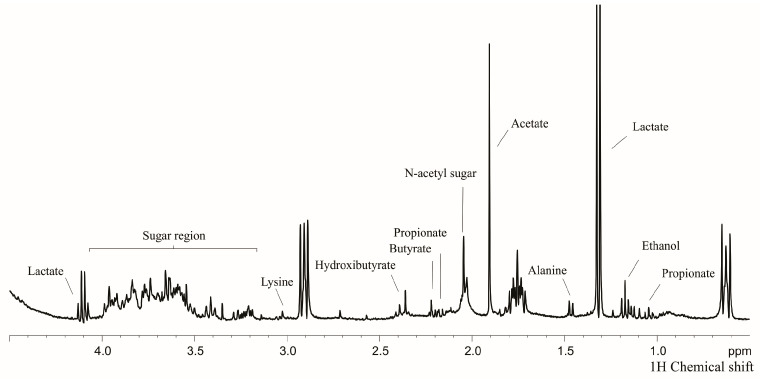
Representative ^1^H-NMR spectrum of the saliva of young children (20 months old): the 0.5 to 4.5 ppm region.

**Figure 2 metabolites-13-00445-f002:**
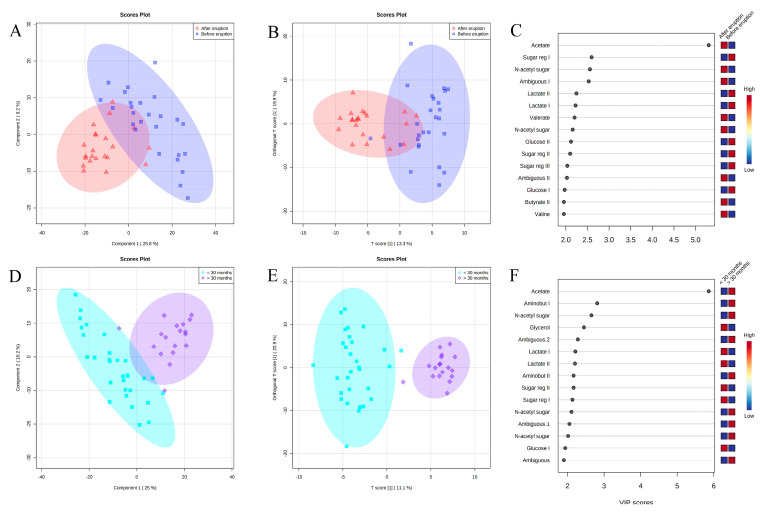
Multivariate analysis of NMR data. (**A**) PLS-DA and (**B**) O-PLS-DA data from children before and after dental eruption; (**D**) PLS-DA and (**E**) O-PLS-DA data from children younger vs. older than 30 months. (**C**,**F**) shows VIP scores for the metabolites that were responsible for differences between the groups.

**Figure 3 metabolites-13-00445-f003:**
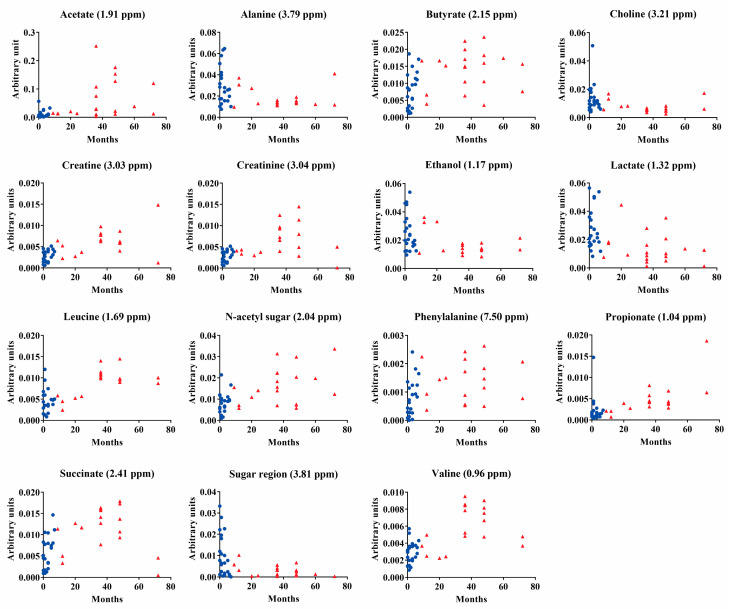
Normalized intensity of the peaks representing the amount of each metabolite (*y* axis) in each child’s saliva, throughout the months of life (*x* axis). Blue and red colors show subjects without and with teeth, respectively.

**Figure 4 metabolites-13-00445-f004:**
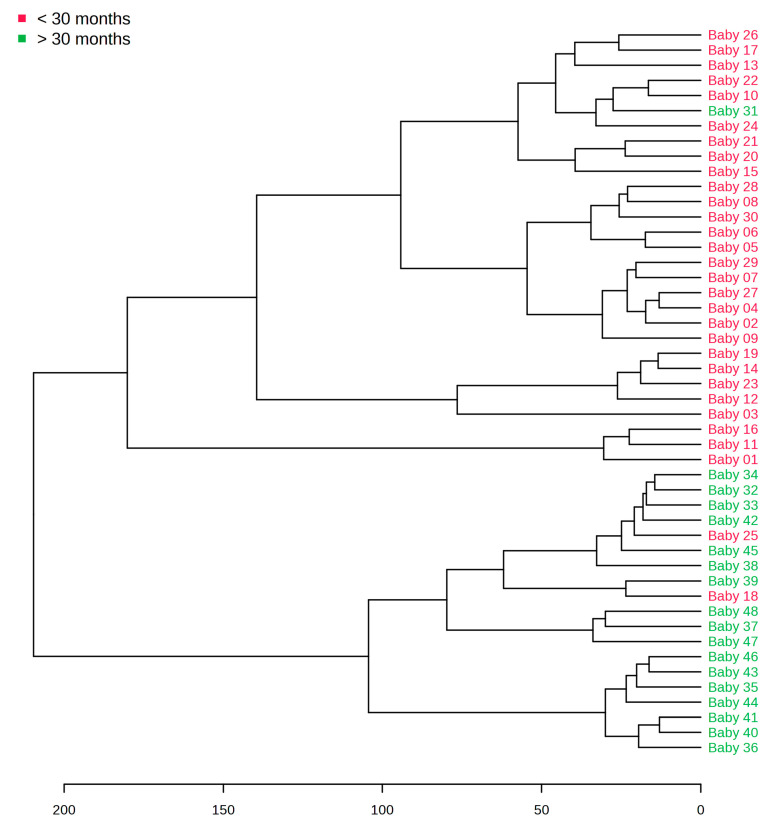
Dendrogram demonstrating the clustering of children < and >30 months of age.

**Figure 5 metabolites-13-00445-f005:**
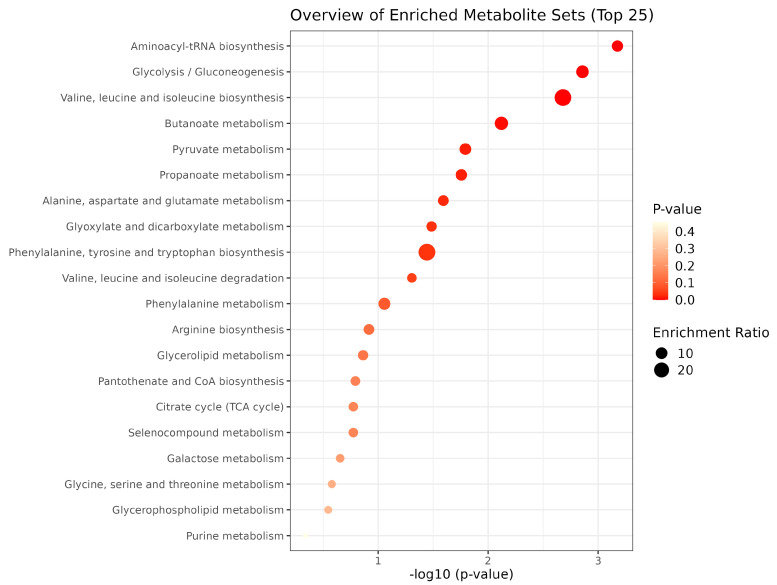
Top 25 metabolic pathways related to the comparison of salivary metabolites from infants < and >30 months old.

**Table 1 metabolites-13-00445-t001:** Metabolites that differ between groups: < vs. >30 months of age.

Metabolites Metabocard	Chemical Shift	>30 Months Mean (95% CI)	<30 Months Mean (95% CI)	*p*-Value
Acetate ^1,2^ HMDB0000042	1.91	11.1 × 10^−3^ (0.4 × 10^−3^–65.0 × 10^−3^)	76.5 × 10^−3^ (3.1 × 10^−3^–251.0 × 10^−3^)	<0.01
Alanine ^2^ HMDB0001310	3.79	25.6 × 10^−3^ (4.0 × 10^−3^–64.0 × 10^−3^)	15.5 × 10^−3^ (8 × 10^−3^–40.0 × 10^−3^)	0.02
Butyrate ^1,2^ HMDB0000039	2.36	7.9 × 10^−3^ (1.0 × 10^−3^–10.0 × 10^−3^)	14.0 × 10^−3^ (3.2 × 10^−3^–14.5 × 10^−3^)	<0.01
Choline ^2^ HMDB0000097	3.51	11.7 × 10^−3^ (3.0 × 10^−3^–50.0 × 10^−3^)	7.0 × 10^−3^ (2.0 × 10^−3^–10 × 10^−3^)	0.04
Creatinine ^1,2^ HMDB0000562	3.03	2.7 × 10^−3^ (0.4 × 10^−3^–8.0 × 10^−3^)	6.8 × 10^−3^ (1.0 × 10^−3^–14.0 × 10^−3^)	<0.01
Ethanol ^1,2^ HMDB0000108	3.61	23.1 × 10^−3^ (0.4 × 10^−3^–50.0 × 10^−3^)	14.9 × 10^−3^ (8.0 × 10^−3^–16.0 × 10^−3^)	0.02
Formate HMDB0000142	8.46	0.07 × 10^−3^ (1.0 × 10^−3^–3.0 × 10^−3^)	0.5 × 10^−3^ (2.0 × 10^−3^–7.0 × 10^−3^)	0.25
Glycerol ^1,2^ HMDB0000131	3.63	38.0 × 10^−3^ (7.2 × 10^−3^–203.2 × 10^−3^)	17.3 × 10^−3^ (6.9 × 10^−3^–36.8 × 10^−3^)	0.03
Lactate ^1,2^ HMDB0000190	1.32	24.3 × 10^−3^ (3.3 × 10^−3^–50.0 × 10^−3^)	11.7 × 10^−3^ (2.0 × 10^−3^–20.0 × 10^−3^)	<0.01
Leucine ^2^ HMDB0000687	1.69	4.1 × 10^−3^ (2.0 × 10^−3^–10.0 × 10^−3^)	10.3 × 10^−3^ (2.2 × 10^−3^–20.0 × 10^−3^)	<0.01
N−acetyl−sugar ^1,2^	2.02	9.1 × 10^−3^ (0.9 × 10^−3^–20.0 × 10^−3^)	21.2 × 10^−3^ (5.0 × 10^−3^–30.0 × 10^−3^)	<0.01
Phenylalanine ^2^ HMDB0000159	7.50	0.3 × 10^−3^ (1.0 × 10^−3^–1.7 × 10^−3^)	1.4 × 10^−3^ (0.3 × 10^−3^–3.0 × 10^−3^)	0.01
Propionate ^2^ HMDB0000237	1.04	2.0 × 10^−3^ (0.3 × 10^−3^–14.0 × 10^−3^)	6.2 × 10^−3^ (0.6 × 10^−3^–18.0 × 10^−3^)	<0.01
Succinate ^2^ HMDB0000254	2.41	5.7 × 10^−3^ (0.8 × 10^−3^–14.0 × 10^−3^)	10.0 × 10^−3^ (0.4 × 10^−3^–17.0 × 10^−3^)	<0.05
Sugar region ^1,2^	3.81	26.5 × 10^−3^ (4.0 × 10^−3^–60.0 × 10^−3^)	15.2 × 10^−3^ (9.0 × 10^−3^–37 × 10^−3^)	<0.01
Trimethylamine ^2^ HMDB0000906	2.98	1.7 × 10^−3^ (0.03 × 10^−3^–5.0 × 10^−3^	4.4 × 10^−3^ (0.6 × 10^−3^–12.0 × 10^−3^	<0.01
Valerate ^1,2^ HMDB0000892	2.15	6.5 × 10^−3^ (1.2 × 10^−3^–13.1 × 10^−3^)	10.6 × 10^−3^ (3.2 × 10^−3^–14.5 × 10^−3^)	<0.01
Valine ^2^ HMDB0000883	0.96	2.7 × 10^−3^ (0.5 × 10^−3^–5.0 × 10^−3^)	6.6 × 10^−3^ (1.4 × 10^−3^–9.0 × 10^−3^)	<0.01
Caproic acid ^2^ HMDB0000535	0.91	4.5 × 10^−3^ (0.8 × 10^−3^–10.0 × 10^−3^)	9.4 × 10^−3^ (4.4 × 10^−3^–14.0 × 10^−3^)	<0.01
Urea HMDB0000294	5.7	1.4 × 10^−3^ (0.3 × 10^−3^–5.0 × 10^−3^)	1.6 × 10^−3^ (0.17 × 10^−3^–5.0 × 10^−3^)	0.72

^1^ Indicates metabolites that presented a statistical difference in VIP score; ^2^ Indicates metabolites that presented a statistical difference in univariate analysis.

## Data Availability

The data are available at https://doi.org/10.17605/OSF.IO/VFXRA.

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
