# Peer review of "NMR-Based Metabolomics Demonstrates a Metabolic Change during Early Developmental Stages from Healthy Infants to Young Children"

_metabolites, 2023, doi:10.3390/metabo13030445_

Round 1

Reviewer 1 Report (Previous Reviewer 2)

There lack of scientific novelty and appropriate study design.
There was a significant ethical problem.

The IRB approval was obtained this year “Brazil (Number 4.712.999, approval data: May 21th 2022).”

However, Metaboanalyst 3.0 was available only four or five years ago.

Currently, only Metaboanalyst 5.0 is available. 

Table 1. Consistent, effective digits should be used. Define ND.

L130 Fasting time was only 15 min. The authors should evaluate the effect of this fasting time on saliva first. 

L226 “Lactate, glycerol, glucose, and sugar region were found in greater amounts in the saliva of patients before 30 months old.” Yes. This is true. However, are the change in the metabolites specific to this study? The authors must prove the specificity of the change in the data.

The authors did not evaluate the linearity between the actual metabolite concentration and signal size. The evaluation of S/N ratio is not enough. The subsequent analyses strongly depend on the linearity between the actual metabolite concentration and signal size. Upper and lower limits also should be provided. The various concentration of identified metabolites should be spiked into each sample and linearity must be evaluated. If the samples were measured in multiple batches, the authors must prepare and confirm the variance of quality control samples. Anyway, the authors did not evaluate the reproducibility of the data.

Author Response

We would like to thank the Editor and Reviewers for considering our manuscript (ID metabolites-2075244), entitled "NMR-based metabolomics demonstrates a metabolic change during early developmental stages from healthy infants to young children" for publication in Metabolites, section “Frontiers in Metabolomics”, Special Issue “Salivary Fingerprint in Metabolomics Era: Potential and Challenges”. The major suggestions were accepted and incorporated into the revised version of the manuscript and some questions were clarified. All reviewers' comments were carefully considered and responded point-by-point below. All changes made to the text are highlighted in track changes. We hope that our answers and changes made in the text have satisfactorily addressed all comments.

Reviewer 1

There lack of scientific novelty and appropriate study design.

R:  We reinforce that there are no studies assessing the physiological changes that occur during infancy related to age and tooth eruption in the literature and we highlighted it in the introduction in our last response. Studies related to early ages are scarce, especially because this is a very difficult sample to be obtained. The introduction and also the study design was deeply reformulated according to reviewer suggestions.

There is limited knowledge related to the saliva profile of infants in the first months of life, mainly due to the difficulty of obtaining samples during this period. (...)

 Although some information is findable in the literature related to the protein content from infants, limited knowledge is available concerning low molecular weight metabolites in infants.”

There was a significant ethical problem. The IRB approval was obtained this year “Brazil (Number 4.712.999, approval data: May 21th 2022).” However, Metaboanalyst 3.0 was available only four or five years ago. Currently, only Metaboanalyst 5.0 is available.

R:  Probably occurred misunderstanding, the approval data was May 14th 2021. The reviewer is right, the version is Metaboanalyst 5.0. This information was updated in the manuscript:

The generated buckets were submitted to multivariate analysis using the Metaboanalyst 5.0 program (www.metaboanalyst.ca).”

Table 1. Consistent, effective digits should be used. Define ND.

R:  The digits were standardized, and the ND (no statistical difference) was replaced by the exact p-values. A note was included in the Table 1:

1Indicates metabolites that presented statistical difference in VIP score; 2 Indicates metabolites that presented statistical difference in univariate analysis.”

L130 Fasting time was only 15 min. The authors should evaluate the effect of this fasting time on saliva first.

R:  Thank you for this comment. There are two important points to clarify. The fasting time in infants must not be similar to older subjects (in general we recommend 1-2h to refrain from oral activities). It would be not ethical recommend this period of breastfeeding infants, therefore we stablish 15 minutes. Besides, we published a study (Letieri et al., 2021) assessing the effect of the mouth cleaning using a gauze moistened with filtered water and we verified that the cleaning was able to remove breastfeeding components. Therefore, we adopted this strategy in the current study. This information was included in the text for readers clarification.

“Mothers were asked to refrain from feeding 15 minutes before saliva collection. For infants, prior to saliva collection their oral mucosa was cleaned, using a gauze moistened with filtered water and after 5 minutes the saliva was collected. For children older than 2 years, it was asked to refrain oral activities for 1 hour prior saliva collection.”

Reference:

LETIERI, A. S. ; FERNANDES, L. B. F. ; ALBARELLO, L. L. ; FONTES, G. P. ; SOUZA, I. P. R. ; VALENTE, A. P. ; FIDALGO, TATIANA K. S. . Analysis of salivary metabolites by Nuclear Magnetic Resonance before and after oral mucosa cleaning of infants in the pre-dental period. Frontiers in Dental Medicine, v. 2, p. 1-9, 2021.

L226 “Lactate, glycerol, glucose, and sugar region were found in greater amounts in the saliva of patients before 30 months old.” Yes. This is true. However, are the change in the metabolites specific to this study? The authors must prove the specificity of the change in the data.

R:  Many metabolites changed. Table 1 shoes the modifications in amino acids, carbohydrates, organic acids, urea, creatinine, and others. We reinforce that we proposed to study a physiologic condition and not a disease, therefore the events that occurred during physiologic maturation, such as gland maturation and tooth eruption, are responsible to change the whole profile of salivary components.

The authors did not evaluate the linearity between the actual metabolite concentration and signal size. The evaluation of S/N ratio is not enough. The subsequent analyses strongly depend on the linearity between the actual metabolite concentration and signal size. Upper and lower limits also should be provided. The various concentration of identified metabolites should be spiked into each sample and linearity must be evaluated. If the samples were measured in multiple batches, the authors must prepare and confirm the variance of quality control samples. Anyway, the authors did not evaluate the reproducibility of the data.

R:  The quantitative metabolomics is focused on identifying and quantifying as many metabolites in a biological sample as possible (Whishart, 2010). The assessments suggested by the reviewer is crucial for quantitative analysis, however we want to clarify that in the current study, our aim was not quantify the metabolites, but assess the abundancies of metabolites, as our previous published studies were performed. Therefore, the strategy is different quantitative metabolomics based on NMR or Mass spectroscopy.  This information was clarified in the text:

“Therefore, the relative abundances of metabolites of studies groups were obtained and subjected to the statistical analysis.”

            Quantitative NMR uses certificated standards of pure metabolites samples with high purity degree and the error margins can be estimated. Usually, a calibration with different concentrations of internal reference (TSP or DSS) is performed and the relationship between peak of pure standards and its concentrations is calculated to obtain assess the linearity. We reinforce that our objective is not to perform a quantitative metabolomic, our strategy was to conduct an untargeted metabolomics and to compare the metabolites abundances of the studies groups, therefore it is not possible to elect a majority peak to do such analyses.

Regarding the reproducibility assessment, as previous mentioned, some samples were acquired twice and also signal/noise obtained. In addition, other measures are adopted to guarantee low variability during the sample preparation and spectra acquisition. All samples were prepared by the same experiment investigator and the data acquisition by another one, according to (Stavarashi et al., 2022). Some analysis was also performed using two different equipment (400 and 500 MHz), demonstrating an acquisition standard.

Wishart DS. Trends in Quantitative Metabolomics. J Biomol Tech. 2010 Sep;21(3 Suppl):S6–7. PMCID: PMC2918081.

Stavarache C, Nicolescu A, Duduianu C, Ailiesei GL, Balan-Porcăraşu M, Cristea M, Macsim AM, Popa O, Stavarache C, Hîrtopeanu A, Barbeş L, Stan R, Iovu H, Deleanu C.Diagnostics (Basel). 2022 Mar; 12(3): 559. PMCID: PMC8947115

Reviewer 2 Report (New Reviewer)

The current manuscript entitled “NMR-based metabolomics demonstrates a metabolic change during early developmental stages from healthy infants to young children” by Liana Bastos Freitas-Fernandes et al. is well written. However, the following points need to be addressed before publication –

1# The concluding remarks is absent in the Abstract section. The authors are suggested to add couple of concluding sentence which will represent the impact of this finding.

2# The font size of the text in the Figure 2 is too small and not clear. The authors are suggested to edit the figure.

3# The manuscript is written in the first person manner (We, Our……). However, a manuscript should be written in the style of third person manner. The authors are suggested to rewrite or edit the entire manuscript.

4# The title of this current manuscript needs to be more specific. The authors are suggested to be specific with teeth eruption stage or at the age of 30 months.

5# The authors have observed a metabolic changes during teeth eruption. Are they pioneer to this finding? If the answer is ‘yes’ then no suggestion is needed. But, if the answer is ‘no’ then what is the novelty of this work? The authors are requested to clarify this issue in their manuscript.

Author Response

We would like to thank the Editor and Reviewers for considering our manuscript (ID metabolites-2075244), entitled "NMR-based metabolomics demonstrates a metabolic change during early developmental stages from healthy infants to young children" for publication in Metabolites, section “Frontiers in Metabolomics”, Special Issue “Salivary Fingerprint in Metabolomics Era: Potential and Challenges”. The major suggestions were accepted and incorporated into the revised version of the manuscript and some questions were clarified. All reviewers' comments were carefully considered and responded point-by-point below. All changes made to the text are highlighted in track changes. We hope that our answers and changes made in the text have satisfactorily addressed all comments.

Reviewer 2

The current manuscript entitled “NMR-based metabolomics demonstrates a metabolic change during early developmental stages from healthy infants to young children” by Liana Bastos Freitas-Fernandes et al. is well written. However, the following points need to be addressed before publication –

R:  We acknowledge for this comment, and we hope that the modifications helped to make the manuscript clearer for readers.

1# The concluding remarks is absent in the Abstract section. The authors are suggested to add couple of concluding sentence which will represent the impact of this finding.

R:  A sentence was included:

“The saliva profile is a result of changes in age and dental eruption and these finding can be useful to monitoring the physiological changes that occur in infancy.”

 2# The font size of the text in the Figure 2 is too small and not clear. The authors are suggested to edit the figure.

R:  We acknowledge for the comment, however this figure is provided automatically by the software and this software (Metaboanalyst 5.0) do not allow change the letter size. The unique alternative to change the letters size is to do it artificially using an editing program (Photoshop or CorelDrow), but we judge that is not a good way to change the size, since it would modify the original data.

3# The manuscript is written in the first person manner (We, Our……). However, a manuscript should be written in the style of third person manner. The authors are suggested to rewrite or edit the entire manuscript.

R:  The text was revised and the first person was changed to third person manner, as reviewer suggested.

4# The title of this current manuscript needs to be more specific. The authors are suggested to be specific with teeth eruption stage or at the age of 30 months.

R:  The title was already changed. Since there are two variables (age and tooth eruption) we opted to maintain the title more unspecific and include the words infants and children.

5# The authors have observed a metabolic changes during teeth eruption. Are they pioneer to this finding? If the answer is ‘yes’ then no suggestion is needed. But, if the answer is ‘no’ then what is the novelty of this work? The authors are requested to clarify this issue in their manuscript.

R:  Thank you for this comment, there are no studies, for the best of our knowledge, that assessed the impact of teeth eruption in salivary metabolomics.

Reviewer 3 Report (New Reviewer)

Manuscript review: NMR-based metabolomics demonstrates a metabolic change during early developmental stages from healthy infants to young children. Thank you for the opportunity to read this manuscript before publication. Here are my comments and recommendations:

Introduction

I find this section well structured, it presents the background and rationale sequentially. At the end of the section is the purpose of the study, which I believe should be a separate paragraph or even a subsection for clarity. But it is not necessary, so I leave this cosmetic change to the Authors' decision.

Methodology

I have no criticisms of this section - I find it correct and clearly organized. In the case of SPSS, the designation in parentheses (publisher, location) is missing.

Results

Figure 3 is too small and therefore illegible. Other than that, I find this section well-written.

Discussion

The discussion is concise and to the point. At the end of the discussion, the limitation subsection is missing, which I strongly recommend adding.

Back Matter

Consider listing specific grant numbers in the Funding subsection - funders may require it.

In conclusion, I consider this manuscript to be very good - better than most papers submitted to journals of this level. Congratulations and wish you more success.

Author Response

We would like to thank the Editor and Reviewers for considering our manuscript (ID metabolites-2075244), entitled "NMR-based metabolomics demonstrates a metabolic change during early developmental stages from healthy infants to young children" for publication in Metabolites, section “Frontiers in Metabolomics”, Special Issue “Salivary Fingerprint in Metabolomics Era: Potential and Challenges”. The major suggestions were accepted and incorporated into the revised version of the manuscript and some questions were clarified. All reviewers' comments were carefully considered and responded point-by-point below. All changes made to the text are highlighted in track changes. We hope that our answers and changes made in the text have satisfactorily addressed all comments.

Reviewer 3

Manuscript review: NMR-based metabolomics demonstrates a metabolic change during early developmental stages from healthy infants to young children. Thank you for the opportunity to read this manuscript before publication. Here are my comments and recommendations:

R:  The authors acknowledge for the reviewer contribution.

Introduction

I find this section well structured, it presents the background and rationale sequentially. At the end of the section is the purpose of the study, which I believe should be a separate paragraph or even a subsection for clarity. But it is not necessary, so I leave this cosmetic change to the Authors' decision.

R:  We followed the reviewer’ suggestion and separate the objective paragraph.

" Therefore, the present study aimed to characterize the salivary metabolomic profile during the early developmental stages of a healthy infant and young children by Nuclear Magnetic Resonance.”

Methodology

I have no criticisms of this section - I find it correct and clearly organized. In the case of SPSS, the designation in parentheses (publisher, location) is missing.

R:  The information was included:

“The Statistical Program SPSS 20.0 (IBM, IL, Chicago, USA)…”

Results

Figure 3 is too small and therefore illegible. Other than that, I find this section well-written.

R:  Thank you for your comment. The figure 3 was changed to better visualization, the size was 12, after change the axis and numbers to size 16 and title to size 18.

Discussion

The discussion is concise and to the point. At the end of the discussion, the limitation subsection is missing, which I strongly recommend adding.

R:  The authors acknowledge for this comment. The limitation of the study was included:

“A limitation of the present study was the study design that was cross sectional, further studies should be conducted considering a longitudinal study where the same infant would be followed after tooth eruption and after 30 months.”

Back Matter

Consider listing specific grant numbers in the Funding subsection - funders may require it.

R:  We included in this session the fundings numbers accordingly.

In conclusion, I consider this manuscript to be very good - better than most papers submitted to journals of this level. Congratulations and wish you more success.

R:  Once again, thank you for the comment, we hope that this manuscript raises interest of readers of metabolomic area.

Reviewer 4 Report (New Reviewer)

This is an interesting study and well-written. There are typos here and there. So, authors should carefully recheck the manuscript. 

Overall, I have no major comments. I wish the samples from the same subject were taken at different time points. Nevertheless, I would like to suggest authors celebrate the conclusion by summarizing the work with possible limitations.  

Author Response

We would like to thank the Editor and Reviewers for considering our manuscript (ID metabolites-2075244), entitled "NMR-based metabolomics demonstrates a metabolic change during early developmental stages from healthy infants to young children" for publication in Metabolites, section “Frontiers in Metabolomics”, Special Issue “Salivary Fingerprint in Metabolomics Era: Potential and Challenges”. The major suggestions were accepted and incorporated into the revised version of the manuscript and some questions were clarified. All reviewers' comments were carefully considered and responded point-by-point below. All changes made to the text are highlighted in track changes. We hope that our answers and changes made in the text have satisfactorily addressed all comments.

Reviewer 4

This is an interesting study and well-written. There are typos here and there. So, authors should carefully recheck the manuscript. 

R:  Thank you for your comment. The text was revised to correct typos.

Overall, I have no major comments. I wish the samples from the same subject were taken at different time points. Nevertheless, I would like to suggest authors celebrate the conclusion by summarizing the work with possible limitations.  

R: We totally agree with this reviewer, this limitation and reflection was made at the end of discussion session.

“A limitation of the present study was the study design that was cross sectional, further studies should be conducted considering a longitudinal study where the same infant would be followed after tooth eruption and after 30 months.”

Round 2

Reviewer 1 Report (Previous Reviewer 2)

The authors provided only references and did not provide any data for reinforces the validity of the data. To obtain reproducible data, the authors must evaluate the linearity between linearity and metabolite concentration since the subsequent statistical analyses strongly depend on the quality of the quantified data. 

Providing only the reference of fasting time is not enough. The authors should evaluate and design the current protocol enough to provide the scientifical meaningful data.

Even though the sample is unique, the authors merely collect samples, ran NMR, and did statistical analyses, without evaluating reproducibility without the evaluating standard of protocol, without confirming the linearity of the data.

To enhance the confident conclusion, the current data and analyses are inadequate.

Reviewer 2 Report (New Reviewer)

The authors have answered all questions. However, the following points must be considered by the journal authority before publishing -

1# The manuscript still contains several 'Our' which represent ignorance of the authors. Need more careful approach.

2# The text of figure is really too small, hard to read. The authors need to find a solution anyhow. But if the current text size meets journal requirement then the reviewer has nothing to say.

This manuscript is a resubmission of an earlier submission. The following is a list of the peer review reports and author responses from that submission.

Round 1

Reviewer 1 Report

The authors present their study of the evolution of the saliva metabolome during an infant's development, with particular emphasis on the difference between the pre- and post-teething metabolome. A potentially interesting topic, but unfortunately far too little is done with the measured data. No attempt was made to check if the differences between the groups was a consequence of teething, gland development, saliva microbiome development or dietary changes. The authors say they collect a great amount of metadata, but use very little of it. The introduction and the discussion are too much a collection of loose sentences and have too little coherence, hence providing little scientific insight.

Some additional comments:

- PCA was mentioned but not shown. Showing the PCA scores and loadings plots and a (O)PLS-DA plot would be more informative than showing both the PLS-DA and OPLS-DA.

- The spectra were normalized, but the authors do not describe exactly how. "... dividing this by the sum of the signal intensities a region of the respective spectra", what region? It was also not mentioned if the data was subjected to scaling (autoscaling, Pareto scaling) prior to multivariate analysis.

-  Table 1 and figure 3 show metabolite peak "intensities"; it is unclear to me what this means? Peak heights? Peak integrals? Bin values? Peak heights should not be used as these are too dependent on the shimming.

- Why was the pre- and post-teething samples chosen to define the groups for the multivariate anlysis, but younger/older than 30 months of age for the univariate analysis?

 - From which part of the spectra were the bins collected? Figure 1 only shows the upfield half of the spectrum, were the downfield parts also used for the statistical analysis?

Reviewer 2 Report

Dr. Liana Bastos Freitas-Fernandes,’s study collected saliva samples from newborns to young children and analyzed these samples using NMR. Unfortunately, there was no scientific novelty. The study design was poorly designed, and the authors were stuck in the NMR world.

Firstly, the measurement aspect of this manuscript is not technically sound.

At least the authors should validate the linearity between the signal and metabolite concentration in the samples. The linear range should be analyzed and presented, and the recovery rate also should be given. How was the quality of the data evaluated? Was the data in this paper reproducible?

Second, the analyses in the clinical study were too poor. For example, how about the fasting condition?

Why was saliva collection volume not measured?

Third, the data analyses were too poor. Run pls-da, list up VIP score. Everyone can reach the possible biomarker. How is the specificity of the change acetate to the phenotype of the interest?